# Polymorphism of the *BMPR*1B Variants for Prolific Traits in the Indonesian Local Ettawah Goat

**DOI:** 10.3390/ani15192781

**Published:** 2025-09-24

**Authors:** Mudawamah Mudawamah, Muhammad Zainul Fadli, Gatot Ciptadi, Fatchiyah Fatchiyah, Mahayu Woro Lestari, Yudith Oktanella, Susiati Susiati, Albert Linton Charles

**Affiliations:** 1Department of Animal Science, University of Islam Malang, Malang 65144, Indonesia; 2Department of Medicine, University of Islam Malang, Malang 65144, Indonesia; 3Department of Animal Science, Brawijaya University, Malang 65145, Indonesia; 4Department of Biology, Brawijaya University, Malang 65145, Indonesia; 5Department of Agroindustry, University of Islam Malang, Malang 65145, Indonesia; 6Department of Veterinary, Brawijaya University, Malang 65151, Indonesia; 7Division of Innovation and ScienceTech Area, Brawijaya University, Malang 65145, Indonesia; 8Department of Tropical Agriculture and International Cooperation, National Pingtung University of Science and Technology, Pingtung 912, Taiwan

**Keywords:** small ruminant, PCR amplification, genotyping, sustainable breeding, agricultural productivity, food security

## Abstract

A polymerase chain reaction (PCR) assay, employing a novel combination of A and G allele primers of the Bovine Morphogenetic Protein Receptor type 1B (*BMPR*1B) gene, revealed significant polymorphism within the Indonesian Local Ettawah Goat (ILEG). Polymorphism analysis classified the goats into three distinct genotypes: homozygous allele A (AA), heterozygous (AG), and homozygous allele G (GG). These findings also indicated a correlation between the presence of the G allele and increased prolificacy and highlighted the potential of *BMPR*1B variants as genetic markers for enhanced reproductive performance and overall livestock improvement.

## 1. Introduction

The increasing impact of climate change and disasters on global food systems reinforces the need to ensure food security from animal protein. Goats, due to their widespread distribution across diverse geographical regions including Asia, Africa, Europe, the Americas, and Australia, and a substantial global population [1,2,3,4,5,6], represent a significant livestock resource. In Indonesia, a Southeast Asian country, the goat population is considerable, with approximately 1.8 million goats spread throughout its 34 provinces. East Java is particularly noteworthy for its high concentration of goats, especially the Indonesian Local Ettawah Goat (ILEG), also known as Peranakan Ettawah, which constitutes the second-largest goat population in rural areas. These goats are primarily raised for dairy and meat using both artificial insemination and natural breeding methods [7].

The Indonesian Local Ettawah Goat (ILEG) is an indigenous Indonesian dual-purpose breed (meat and milk production) exhibiting promising genetic potential for reproductive traits, particularly prolificacy, with the capacity to produce multiple offspring per parturition; however, this inherent potential remains underexploited. Surveys conducted across 19 breeding villages revealed that the litter size of ILEG ranges from 1.42 to 2.12 offspring, averaging 1.72 (<2 offspring per parturition), with a corresponding phenotypic variation of 0.53 and an estimated breeding value (EBV) ranging from 1.48 to 1.74 [8]. These observations corroborate that the on-farm productivity of ILEG has yet to attain its full genetic capacity, indicative of low reproductive efficiency and a significant disparity (productivity gap) between theoretical potential and realized performance. Consequently, the present investigation is highly pertinent for informing practical strategies to enhance the performance of ILEG meat and milk production systems on smallholder farms in East Java, with a particular focus on improving prolificacy.

Prolificacy, commonly measured by litter size, is a key reproductive indicator that directly influences small ruminant productivity and the profitability of farming systems. Defined as the number of offspring per parturition, litter size reflects a goat’s reproductive potential and overall contribution to livestock productivity. This trait exhibits moderate heritability, enabling the transmission of fecundity-related genes across generations and making genetic improvement through marker-assisted selection a promising strategy. Advances in molecular biology have facilitated the identification of candidate genes associated with fertility in small ruminants, including bone morphogenetic protein receptor type 1B (*BMPR1B*), *BMPR15*, *GDF9*, *INHA*, and *INHB* [9]. Among these, *BMPR1B* is of particular interest due to its established role in ovulation mechanisms and litter size, with genetic polymorphisms shown to enhance both ovulation rate and fertility [10,11].

Molecular studies have examined the genetic basis of prolificacy in various goat breeds, revealing diverse responses to *BMPR*1B variants [12,13,14,15,16,17,18]. For example, positive associations between *BMPR*1B and prolificacy have been reported in Black Bengal [19,20], Beetal, and Teddy goats [21], whereas inconsistent results have been observed in tropical breeds such as Malabary, Attappady Black [22], and Tibetan Cashmere [23]. For instance, the genetic basis of prolificacy in Malabari and Attappady goats, using polymerase chain reaction (PCR) and sequencing methodologies on limited sample size (*n* < 50), identified the presence of the heterozygous AG genotype; however, no statistically significant association was observed between this genotype and prolificacy [22]. Similarly, a PCR-RFLP analysis of 216 Tibetan Cashmere goats identified single nucleotide variations (SNVs) in BMPR1B but no polymorphic restriction fragment length patterns. The absence of detectable polymorphism may reflect limited genetic diversity within these populations, breed-specific differences in *BMPR*1B allele frequencies, and/or methodological differences in polymorphism detection.

Variation in allele distribution, particularly the G and A alleles of *BMPR*1B, is a key factor influencing these outcomes. These variations likely stem from structural changes in the *BMPR*1B receptor, which affect the TGF-B signaling pathway involved in folliculogenesis [8]., although this relationship is breed-dependent. Therefore, investigating genetic polymorphism is crucial for identifying associations with prolific traits and understanding the frequency distribution of genes within a population.

Limited molecular research has explored the genetic basis of prolificacy in the Indonesian Local Ettawah Goat (ILEG), with previous studies reporting associations with the GDF9 gene [7] as well as *BMP*15, *BMPR*1B, and *KISS*1 [24,25,26]. For example, two genotypes were identified within exon 1 of the *BMPR*1B gene using a PCR-RFLP approach, although a definitive causal relationship with prolificacy was not established [25]. Building on these findings, the present study aimed to characterize haplotype diversity and genotype frequency distributions of the *BMPR*1B gene in ILEG populations in East Java and to evaluate their association with prolificacy. The outcomes are expected to provide useful molecular markers for ILEG breeding programs, supporting improvements in reproductive efficiency and overall productivity.

## 2. Materials and Methods

### 2.1. Data Collection

The descriptive study used samples collected from ILEG parents at various village breeding centers of East Java, including Malang I (Ampel Gading), Malang II (Lawang and Singosari), Blitar, Gresik, Lumajang, Nganjuk, Trenggalek, and Banyuwangi. The sample comprised 385 goats without recorded prolificacy data and 75 goats with documented prolificacy records. Prolificacy data were obtained through direct and individual interviews with breeders at their goat housing facilities. Breeders with a herd size of 3–5 animals were selected to enhance the accuracy of their recall regarding individual litter sizes. The recorders employed in the study consisted of 2–4 undergraduate animal husbandry students who received prior training from the researchers. To avoid biased data, an internal cross-validation was implemented. This involved comparing farmers’ reported data with directly observable physiological characteristics of the animals at the time of data collection, including the number of kids suckling, the age of the offspring, and the reproductive status of the does.

### 2.2. Isolation of DNA

Blood samples were collected from dams using Terumo needles and syringes and immediately transferred into EDTA-coated tubes (Onemed, Surabaya, Indonesia). During transport to the laboratory, samples were stored in an icebox and subsequently either processed immediately or preserved at −20 °C until further use. Genomic DNA was extracted using a modified salting-out method based on [27]. The extraction procedure employed standard laboratory equipment, including a water bath (Yamato (Harumi Triton Square Office Tower Y 36F, 1-8-11 Harumi, Chuo-ku, Tokyo 104-6136, Japan), vortex mixer ((Genie= scientific industries inc. 80 Orville Drive, Suite 102, Bohemia, New York, 11716, USA), centrifuge (Andreas Hettich GmbH Föhrenstraße 12, 78532 Tuttlingen, Germany)),) spectrofotometer UV vis Genesys 10 (Thermo Electron Scientific Instruments LLC, Madison, WI, USA), and refrigerated microcentrifuge Micro 22R (Andreas Hettich GmbH Föhrenstraße 12, 78532 Tuttlingen, Germany)).

The extraction process involved repeated red blood cell (RBC) lysis, cell lysis, organic solvent extraction, DNA precipitation, and resuspension. Briefly, 0.5 mL thawed blood was transferred into a sterile tube and mixed with 1 mL RBC lysis buffer, vortexed, and incubated for 5 min at room temperature, followed by centrifugation at 3000 rpm for 10 min at 4 °C. The supernatant was discarded, and the pellet was subjected to 3–5 additional cycles of RBC lysis until a white pellet was obtained. The pellet was then resuspended in 300 µL cell lysis buffer, vortexed for 5 min, and incubated at 37 °C for 1 h. After cooling to room temperature, 600 µL of phenol:chloroform:isoamyl alcohol (25:24:1, *v*/*v*) was added, vortexed, and centrifuged at 10,000 rpm for 10 min. The aqueous phase was transferred to a new sterile tube and re-extracted with chloroform:isoamyl alcohol (24:1, *v*/*v*), followed by centrifugation at 10,000 rpm for 10 min.

The resulting supernatant was mixed with 20 µL of 10 M ammonium acetate and 1 mL cold absolute ethanol to precipitate DNA, which was then collected by centrifugation at 10,000 rpm for 10 min at 4 °C. The DNA pellet was washed with 70% ethanol, centrifuged, air-dried at 37 °C, and resuspended in 50 µL TE buffer (pH 7.6). DNA samples were stored at −20 °C for long-term preservation.

DNA quality and concentration were assessed using a UV–Vis spectrophotometer by measuring absorbance at 260 and 280 nm. The A260/A280 ratio ranged from 1.53 to 1.92, and the final DNA yield was 60–150 ng/µL.

### 2.3. PCR Amplification

Primer was order from Integrated DNA Technology Pte. Ltd.-Singapore. Polymerase chain reaction (PCR) was performed using the following reaction mixture: 2 μL ddH2O, 1 μL forward primer, 1 μL reverse primer, 5 μL Promega GoTaq Green Master Mix (Promega cat number M7122), and 1 μL DNA template. The PCR cycling conditions consisted of an initial denaturation step at 94 °C for 3 min, followed by 35 cycles of denaturation at 94 °C for 30 s, annealing at 52 °C for 30 s, and extension at 72 °C for 30 s. A final extension step was performed at 72 °C for 10 min. Two primer sets were used in this study to amplify the BMPR1B gene: Allele G primer set and Allele A primer set [19], as detailed in Table 1.

Table 1 details the nucleotide sequences of *BMPR*1B primers targeting the A and G alleles used in this study, along with the corresponding A/G notation for band identification on PCR results. *BMPR*1B, a serine/threonine protein kinase receptor, belongs to the protein kinase superfamily. STRING database analysis, integrating data from database searches, text mining, experimental evidence, and homology predictions, suggests several functional partners of *BMPR*1B, with confidence scores ranging from 0.983 to 0.999. The predicted interactions between BMPR1B and these potential partners are illustrated in Figure 1.

### 2.4. DNA Sequencing

The PCR amplicons were submitted to 1st BASE (PT Genetika Science, Jakarta, Indonesia) for Sanger sequencing. The resulting sequence data was visualized as chromatograms, with each nucleotide base identified by a distinct color: guanine (G) in black, cytosine (C) in blue, adenine (A) in green, and thymine (T) in red.

### 2.5. Polymorphism

Polymorphism was performed on dams with both recorded and unrecorded prolificacy data. The frequencies of phenotypes, genetic group, and alleles were then analyzed based on the polymorphism results for both groups of dams. To assess differences in the frequencies of the AA, GA, and GG genetic groups.

### 2.6. Statistical Analysis

PCR data for the *BMPR*1B alleles (A and G) were classified into three genotypes: GG, GA, and AA. These genotypes were then combined with the mean prolificacy values of each ewe to assess associations. Data normality was evaluated using the Shapiro–Wilk test to determine the suitability of analysis of variance (ANOVA). For data that failed to satisfy normality assumptions, nonparametric tests (Kruskal–Wallis and Mann–Whitney) with Bonferroni correction were applied. Descriptive statistics, including mean, median, standard deviation, and interquartile range (IQR), were used to summarize the data. The IQR, defined as the difference between the 75th percentile (Q3) and the 25th percentile (Q1), was reported as a measure of data dispersion. All statistical analyses were performed in Python version 3.10 using the SciPy library.

## 3. Results

### 3.1. PCR Results

PCR analysis was conducted on goat genomic DNA to determine the presence of A and G alleles of the *BMPR*1B gene. Two primer sets were utilized: one specific to the A allele, generating a 1100 bp amplicon (Figure 2A), and the other specific to the G allele, generating a 100 bp amplicon (Figure 2B). Genetic groups were assigned based on the presence or absence of these amplicons; for instance, samples exhibiting amplification only at 1100 bp were designated as homozygous for A allele (AA) whereas, samples exhibiting amplification only at 100 bp were designated as homozygous for G allele (GG). Samples demonstrating amplification at both 1100 bp and 100 bp were identified as heterozygous (AG).

### 3.2. Haplotype and Sequencing

Sequencing of the *BMPR*1B allele A in 19 samples revealed no consistent polymorphisms. Any observed variations were interpreted as individual polymorphisms specific to a single sample. Individual polymorphisms identified in the sequencing results are summarized in Table 2 and Table 3.

Sequence analysis of the *BMPR*1B amplicons (*n* = 19) revealed nucleotide frequency variations at specific positions, as summarized in Table 2. Specifically, at positions 133 and 141, adenine (A) was the most frequent nucleotide, observed in 63% and 68% of the sequences, respectively, while cytosine (C) was present in 37% and 32% of sequences. Position 135 showed a high frequency of guanine (G) at 95%, with thymine (T) detected in only 5% of the sequences. Similarly, positions 137 and 143 exhibited a high frequency of guanine (G), at 95% and 63%, respectively, with adenine (A) detected in only 5% and 37%. At position 136, adenine (A) was highly frequent (95%), with guanine (G) detected in the remaining 5%. Finally, position 148 exhibited a thymine (T) frequency of 63%, with guanine (G) observed in 37% of the sequences.

Table 3 presents the results of sequencing analysis aimed at identifying nucleotide variation and corresponding amino acid residue changes within the *BMPR*1B gene across a 133–150 nucleotide sequence. At nucleotide positions 133 and 135, variation from adenine (A) to cytosine (C) and guanine (G) to thymine (T), respectively, resulted in a change in the methionine to leucine at location 45, due to the codon change from AUG to CUU. Similarly, at positions 136 and 137, variation from adenine (A) to guanine (G) and guanine (G) to adenine (A) led to a change in the serine to aspartic acid at amino acid location 46, resulting from the codon change AGC to GAG. At amino acid position 47, nucleotide changed from A to C at nucleotide sequence 141, with the corresponding codon change from CCA to CCC, did not result in an amino acid change (proline). A nucleotide changed from G to A at nucleotide position 141, leading to the codon change from AGA to AAA at amino acid location 48, resulting in a change from arginine to lysine. Finally, a variation from thymine (T) to guanine (G) was observed at nucleotide position 148 and amino acid location 50, resulting in a change from the amino acid leucine to valine, reflected by the codon change from TTA to GTA.

Haplotype diversity was high (0.9415), with 15 distinct haplotypes identified (Table 4). Haplotypes were identified using PCR sequencing of the *BMPR*1B gene allele A, which is known to play a crucial role in reproductive traits. This high level of genetic diversity was attributed to geographical differences among the breeding villages. An exception was haplotype 3, which exhibited genetic similarity across the Malang I (animal code AG), Banyuwangi (BW), and Gresik (GR) breeding villages. Additionally, genetic diversity was observed even among different individuals within the same village. For example, individuals from Malang I (animal code AG) displayed four different haplotypes (2, 4, 6, and 12), while individuals from Banyuwangi (BW) showed haplotypes 5, 8, 10, and 13, and individuals from Gresik (GR) had haplotypes 1 and 9. The specific positions of each haplotype are detailed in Table 4.

### 3.3. BMPR1B Polymorphism of Does with Documented Prolificacy Records

PCR analysis of 73 goats for the *BMPR*1B gene variants identified three distinct genetic groups: homozygous GG, heterozygous AG, and homozygous AA. Each genetic group presented a specific mean prolificacy of does (Table 5).

Table 5 presents the results of polymerase chain reaction (PCR), performed in duplicate using primer sets specific to both the A and G alleles of the *BMPR*1B gene. It showed that homozygous A individuals (AA), identified by amplification with only the *BMPR*1B Allele A primer set, comprising 11 individuals (genotype frequency = 0.15) and exhibited an average prolificacy of 1.49 offspring per kidding per goat.

Table 5 presents the results of polymerase chain reaction (PCR), performed in duplicate using primer sets specific to both the G and A alleles of the *BMPR*1B gene. The homozygous A allele individuals (AA), identified by amplification with only the *BMPR*1B allele A primer set, comprising 11 individuals (genotype frequency = 0.15) and exhibited an average prolificacy of 1.49 offspring per kidding per goat.

Moreover, heterozygous individuals (GA), identified by amplification with both *BMPR*1B Alleles A and G primer sets, comprised 51 individuals (genetic group frequency = 0.70) and exhibited an average prolificacy of 1.60 offspring per kidding per goat. In addition, homozygous variants individuals (GG), identified by amplification with only the BMPR1B allele G primer set, comprised 11 individuals (genotype frequency = 0.15) and exhibited an average prolificacy of 1.89 offspring per kidding per goat.

The results of the descriptive analysis showed that the average prolificacy of GA, GG, and AA individuals is summarized in Table 6. The average number of offspring per kidding per goat for homozygous GG, heterozygous GA, and homozygous AA was 1.89 ± 0.35, 1.60 ± 0.52, and 1.49 ± 0.58, with medians of 2.00, 1.50, and 1.40, and IQRs of 0.00, 1.00, and 0.84, respectively. Meanwhile, ewes with heterozygous GA and homozygous AA had average prolificacy values of 1.60 and 1.49 with medians of 1.50 and 1.40, standard deviations of 0.52 and 0.58, and IQRs of 1.00 and 0.84 offspring per kidding per goat.

The Shapiro–Wilk test indicated that the GG, GA, and AA genotypic groups deviated significantly from normality (*p* ≤ 0.05). As the data did not satisfy the assumptions required for analysis of variance (ANOVA), non-parametric methods were applied. Overall differences among groups were evaluated using the Kruskal–Wallis test, with pairwise comparisons performed using the Mann–Whitney test. This analytical strategy is appropriate for non-normally distributed data and provides more robust statistical inferences.

Boxplots and dot plots of prolificacy across the GG, GA, and AA genotypic groups (Figure 3) further corroborated these findings. The GG group displayed a notably homogeneous distribution, with most values tightly clustered around 2.0, yielding an interquartile range (IQR) approaching zero. In contrast, the GA and AA groups exhibited greater variability, as evidenced by their higher IQR values.

The results of the median comparison and pairwise tests among the three genotypic groups (GG homozygous, GA heterozygous, and AA homozygous) for prolificacy are summarized in Table 7. Pairwise Mann–Whitney analyses revealed significant differences between GG and GA (*p* = 0.029) and between GG and AA (*p* = 0.040), indicating that GG homozygotes generally exhibit higher prolificacy than the other groups. In contrast, no significant difference was observed between GA and AA (*p* = 0.508), suggesting comparable prolificacy levels in these genotypes.

The GG genotypic group exhibited the highest median prolificacy, with significant increases of 33.3% and 42.9% relative to the GA and AA groups, respectively. By contrast, the difference between GA and AA was minimal (7.1%), indicating comparable prolificacy patterns. Collectively, these findings demonstrate that the GG genotype is consistently associated with higher prolificacy, whereas the GA and AA genotypes exhibit similar reproductive performance.

### 3.4. Polymorphism Results Without Recorded Prolificacy Data

The distribution of genotype frequencies across seven breeding villages is summarized in Table 8. Polymorphism of 385 goats with unrecorded prolificacy data for the BMPR1B gene revealed three distinct genotypes: homozygous Allele A (AA), heterozygous (AG), and homozygous Allele G (GG). The observed genotype frequencies and corresponding goat numbers were as follows: AA = 0.04 (53 goats), AG = 0.82 (316 goats), and GG = 0.14 (16 goats). Based on these genotype frequencies, the estimated frequency of the A allele was 0.55, while the estimated allele frequency of the G allele was 0.45.

Table 9 presents the distribution of genotypes, including the number of heterozygous, homozygous allele G, and homozygous allele A individuals, and their corresponding genotype frequencies per breeding village across several breeding village locations in East Java Province, Indonesia. Polymerase chain reaction (PCR) analysis revealed that eight villages (Malang I, Malang II, Blitar, Gresik, Lumajang, Nganjuk, Trenggalek, and Banyuwangi) exhibited a high frequency of the heterozygous genotype (GA), which ranged from 73.91% to 94.74%.

The frequency of the homozygous for allele G (GG) ranged from 5.26% to 20.63% in these villages, with the exception of Lumajang. In addition, the homozygous AA exhibited the lowest frequency, ranging from 0% to 6.38%, with the exception of Lumajang. In Lumajang breeding village, the homozygous GG had the lowest frequency (10.87%), while the frequency of the homozygous AA was 15.22%.

Moreover, the results of the pairwise comparison test using Mann–Whitney with Bonferroni correction showed that the comparison between the GA (Heterozygous) and GG (Homozygous G) groups was significantly different, with a *p*-value of 0.00093 (<0.0167). Similarly, the comparison between GA and AA (Homozygous A) was also significant with a *p*-value of 0.00086 (<0.0167). In contrast, the comparison between GG and AA did not show a significant difference, with a *p*-value of 0.038 (*p* < 0.0167 after Bonferroni correction).

Figure 4 illustrates the distribution of these genetic group frequencies among the different breeding villages. The AA genetic group exhibited the lowest frequency across all breeding villages, with Malang I, Blitar, Gresik, and Nganjuk displaying a frequency of 0. Conversely, the heterozygous genotype displayed the highest frequency across all breeding villages.

## 4. Discussion

Figure 2 shows that PCR amplification yielded 100 bp fragments for allele G and 1100 bp fragments for allele A, corresponding to *BMPR*1B gene variants. These results confirm that the primers effectively discriminated between allelic forms, enabling reliable identification of polymorphisms within the population. Samples exhibiting only the 1100 bp band with the allele A primer were classified as AA, those showing only the 100 bp band with the allele G primer as GG, and those displaying both bands as GA.

Table 2 and Table 3 summarize the significant variations in the nucleotide sequences at positions 133, 135, 136, 137, 141, 143, and 148. These variations include transitions of Adenine (A) to Cytosine (C) (A-C), Guanine (G) to Thymine (T) (G-T), Adenine (A) to Guanine (G) (A-G), Guanine (G) to Adenine (A) (G-A), Thymine (T) to Guanine (G) (T-G). Notably, A to C transition (A-C) occurred at positions 133 and 141. G to T transitions (G-T) occurred at positions 135, an Adenine (A) to Guanine (G) transition was observed at position 136, Guanine (G) to Adenine (A) was observed at positions 137 and 143, A to C transition occurred at 141, and a Thymine (T) to Guanine (G) transition (T-G) was observed at position 148. The frequencies of these nucleotide variations varied sharply, from 5% to a high 37%, with an average incidence of 22.57% within the analyzed (ILEG) population. However, variations at positions 133, 141, 143, and 148 exceeded this average, exhibiting 37% and 32% frequencies for the A to C, G to A, T to G and A to C variations, respectively, which highlights the prevalence of these variations and their potential impact. Sequencing analyses revealed that such variation patterns in goats are prevalent and corroborates findings in other small ruminant species [28,29]. Correspondingly, distinct variations are reported [30] at position 773 (G to C) in Assam hill goats utilizing sequencing with the *BMPR*1B primer. Furthermore, a mutation at position 109 (A to G) was documented in sheep due to crossbreeding between Merino and Garut breeds [10]. A BMPR1B variations at position 159 (A to G) has also been reported in Mehraban sheep [31]. These findings underscore the significance of these nucleotide variations in understanding genetic variations and potential breeding strategies in small ruminants.

Table 2 and Table 3 show three variants in the *BMPR*1B gene in ILEG, located at 133–148, of 45–50 amino acid locations. At the 45th location, there was a change in the methionine amino acid to leucine (AUG to CUU); at the 46th location, there was a change in the serine amino acid to aspartic Acid (AGC to GAC); at the 47th residue, there was no change in the amino acid proline, but there was a change in the nitrogen base compound (CCA to CCC). At the 48th location, there was a change in amino acid residue from arginine to lysine (AGA to AAA); and at the 50th location, there was a change in amino acid residue from leucine to valine (TTA to GTA).

These amino acid variations resulted in different amino acids, yet failed to affect the classification of the amino acid group (essential or non-essential) except at 48th amino acid location there was change from non-essential amino acid to essential amino acid (arginine to lysine). Specifically, variations at location 45 and 50 also resulted in amino acids that retained the properties of the original essential group. The protein resulting from variations at location 46 produced a non-essential group, while location 48 produced an essential group. Reportedly [32], the essential amino acids were methionine, lysine, threonine, leucine, isoleucine, valine, phenylalanine, tryptophan, cysteine, arginine, and histidine, while non-essential amino acids were alanine, aspartate, asparagine, glutamine, glutamate, cysteine, tyrosine, serine, proline, and glycine.

Table 4 show the haplotype diversity of ILEG in the East Java region of Indonesia had a high category of 0.9415, and 94.5% haplotype was recorded [33] between lineage A and B based on analysis of the mitochondrial D-loop region in goats in the Yangtze River region, China. Moreover, high haplotype diversity in seven goat breeds in Russia was similarly reported [34] at 0.843–1; in addition, a study [35] claims that haplotype diversity in seven goat breeds in China ranged from 0.943–1000 with a high category, which corroborated the finding of high genetic diversity in East Asia goats [36].

Table 5, Table 6 and Table 7 and Figure 3 highlight the genetic group frequencies within the population, indicating that AA, AG, and GG were present at 0.15, 0.70, and 0.15, respectively. The BMPR1B gene is connected to two distinct alleles: Allele A, which does not correlate with high fecundity, and the potent Allele G, regarded as a prolific carrier gene. Our PCR analysis, utilizing primers specific to both alleles, confirmed the existence of AA, AG, and GG genotypes in the sample.

Based on statistic descriptive and analysis statistics of prolificacy results AA genetic group yielded an average of 1.49 ± 0.48 offspring; GA produced 1.60 ± 0.52 whereas GG averaged 1.89 ± 0.58, median 33.30–42.90% higher than GA and AA. Significant differences were observed between GG and GA, as well as between GG and AA, whereas no difference was detected between GA and AA. This was supported by the IQR of 0.00 in the GG group, indicating a homogeneous distribution of prolificacy, in contrast to the higher IQR values (0.84–1.00) observed in GA and AA, which reflected greater variability. These findings suggest that individuals with the GG genotype exhibit consistently high reproductive performance, a desirable trait in productivity-related contexts. By comparison, the phenotypic diversity observed in GA and AA may reflect both genetic heterogeneity and environmental influences. For breeding programs, the stability of the GG genotype underscores its potential value in improving prolificacy, while the variability in GA and AA highlights the need for targeted management strategies to optimize their productivity. Collectively, these results indicate that the GG genotype is strongly associated with enhanced prolificacy, whereas GA and AA genotypes display comparable reproductive performance.

This evidence is supported by findings that Black Bengal goats had average litter sizes of 2.7, 3.04, and 3.11 for AA, AG, and GG genotypes, respectively [20]. The prolificacy of ILEG is lower than that of Black Bengal goats, which may be attributed to genetic background differences. Specifically, the frequency of the BMPR1B variant in ILEG (27.15%) is lower than that reported for Black Bengal goats (40%) [20]). Although different genotyping methods were applied (dual-primer PCR in the present study and PCR-RFLP in Black Bengal goats) both approaches targeted the same BMPR1B SNP associated with prolificacy. Thus, the genotype frequencies obtained here are suitable for direct comparison and highlight genetic differences between the two populations. Additionally, research on local Bangladeshi sheep that employed the same genomic primers reported averages of 1.19 ± 0.07, 1.44 ± 0.09, and 2.13 ± 0.09 for the ++, +B, and BB genotypes, respectively [37]. Notably, numerous studies consistently reveal that heterozygous genotypes often outperform AA homozygous genotypes in terms of prolificacy [38].

Variations in reproductive performance among ILEG and Black Bengal goats with GG, GA, and AA genotypes reflected differences in prolificacy, largely attributable to breed-specific genetics and the polymorphism detection methods employed. Genetics play a critical role in maximizing productivity in these valuable livestock populations [39,40]. The present findings in ILEG are consistent with those reported for Black Bengal, Beetal, and Teddy goats, where *BMPR*1B variants show a positive association with prolificacy. By contrast, studies in Malabary, Attappady [22], and Tibetan Cashmere [23] goats have not demonstrated a clear association, likely due to limited genetic variation, small sample sizes, methodological differences, or environmental influences, which were beyond the scope of the present study. These comparisons underscore the breed-specific influence of *BMPR*1B on prolificacy and highlight the value of molecular screening in optimizing selection strategies.

Genotyping of 385 does of ILEG (Table 8) revealed critical insights into the *BMPR*1B gene within the breeding villages. The allele frequencies for A and G were 0.55 and 0.45, respectively. Moreover, the genetic group frequencies were 0.04 for the AA, 0.82 for the GA, and 0.14 for the GG. These findings underline the very low frequency of homozygous AA, emphasizing that high productivity traits are desirable and important criteria for selecting ILEG in rural breeding farms. The very low frequency of AA also correlated negatively with productivity, especially the number of kids per litter, meaning that the higher the livestock productivity, the lower the frequency of homozygous AA. Furthermore, this aligns with the research by Zhao et al. [41], which supports the notion that positive selection for highly prolific traits in Guizhou Black goats and Meigu goats has significantly enhanced the proliferation of high-performing individuals while simultaneously reducing the prevalence of lower-performing traits. This evidence strongly advocates for the strategic focus on prolificacy to improve breeding outcomes and overall livestock performance.

In village breeding populations, heterozygous (GA) individuals represented the largest proportion (Table 7; Figure 3). The frequency of the GA genotype was significantly higher than that of either homozygous group (GG or AA). On average, the AA genotype was the least common, occurring at a frequency 231% lower than GG, although this difference was not statistically significant. This distribution pattern suggests effective positive selection for prolificacy traits within the village breeding goat populations. Despite the lack of formal recording practices, farmers relied on observable traits—such as appearance, size, health, and fertility—and their own experiences to select parent candidates for ILEG. This intuitive selection process aligned perfectly with the higher prevalence of heterozygous GA phenotypes, underscoring that ILEG was crucial in promoting long-term genetic diversity within livestock populations. Local customs, such as the exchange of bucks among breeders, supported this diversity of ILEG. These findings support the idea that the advantages conferred by heterozygous GA arise from adaptive mutations, highlighting the potential for sustainable breeding practices that capitalize on genetic diversity [42,43].

## 5. Conclusions

The study of the BMPR1B gene in Indonesian Local Ettawah Goat (ILEG) revealed significant insights into the genetic basis of prolificacy, with potential applications for enhancing livestock breeding programs. The presence of the allele G correlated with increased fecundity, highlighting BMPR1B’s utility as a genetic marker for reproductive performance. Polymorphism analysis identified three distinct genetic groups (AA, AG, and GG) and their respective frequencies in ILEG populations, both with and without recorded prolificacy data. The heterozygous genotype (GA) was the most prevalent in village breeding goat populations, suggesting successful, positive selection for prolific traits, even in the absence of formal recording practices. The homozygous GG genotype was associated with higher fecundity, and the elevated frequency of allele G underscores the relevance of incorporating prolificacy traits into breeding selection strategies. Accordingly, the breeding program prioritized GG does, while AG bucks were retained to preserve genetic diversity within the herd. Highly prolific individuals were identified through PCR-based polymorphism analysis, enabling the establishment of a core breeding group. Based on these findings, the implementation of sustainable breeding strategies is recommended for ILEG using dual-primer PCR genotyping to efficiently screen for GG and AG individuals, which could increase average prolificacy by an estimated 15–20%.

## Figures and Tables

**Figure 1 animals-15-02781-f001:**
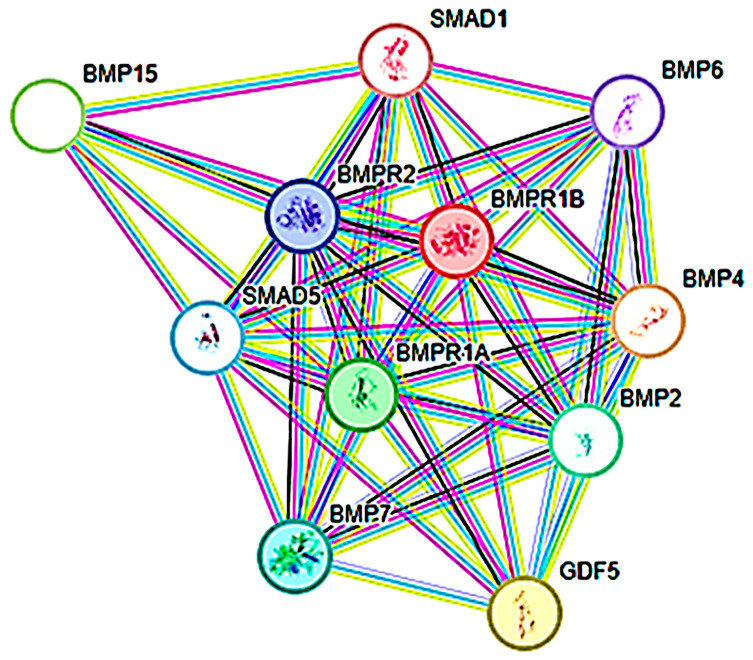
The relationship between *BMPR*1B and primers. *BMP* = bone morphogenetic protein, *GDF* = growth differentiation factor, *MAD* = Small Mothers against decapentaplegic, *BMPR* = bone morphogenetic protein receptor.

**Figure 2 animals-15-02781-f002:**
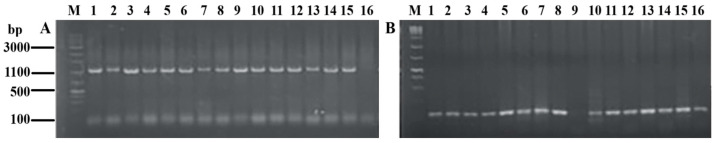
PCR with *BMPR*1B Allele A (**A**) and *BPMR*1B Allele G (**B**); samples in wells no. 1–8 and no. 10–15 had notation AG; well no. 16 had notation GG (**A**); well no. 9 had notation AA (**B**).

**Figure 3 animals-15-02781-f003:**
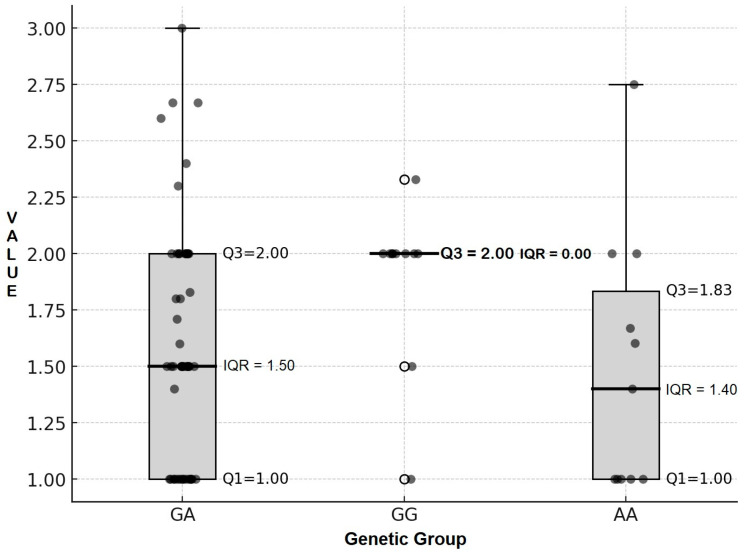
The prolificacy expression across GG, GA, and AA genetic groups.

**Figure 4 animals-15-02781-f004:**
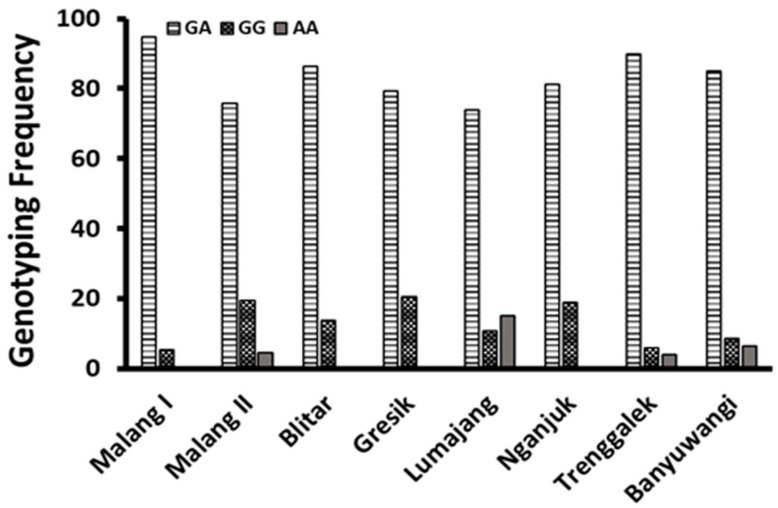
The distribution of genetic group frequency (GA, GG and AA) in several breeding villages of East Java.

**Table 1 animals-15-02781-t001:** Nucleotide sequences of primer BMPR1B.

Type	Nucleotide Sequence	Symbol
*BMPR*1B Allele A	R 5′-gctggttccgagagacagaaatatatca-3′	A
	F 5′-ccccgtccctttgatatctgcagcaatg-3′	
*BMPR*1B Allele G	R 5′-gtcgctatggggaagtttggatgggaa-3′	G
	F 5′-atgttttcatgcctcatcaacaccgtcc-3′	

**Table 2 animals-15-02781-t002:** Nucleotide and Frequency of individual ILEG.

Number of Nucleotide Sequences	*n*	Nucleotide	Frequency (%)	Nucleotide	Frequency (%)
133	19	A	63	C	37
135	19	G	95	T	5
136	19	A	95	G	5
137	19	G	95	A	5
141	19	A	68	C	32
143	19	G	63	A	37
148	19	T	63	G	37
Average			77.43		22.57

**Table 3 animals-15-02781-t003:** Nucleotide Variant and Amino Residue Change.

CDS Position(s)	REF→ALT (nuc)	Codon Change	HGVS.p (AA Change)	Effect (SnpEff)	Impact
133 and 135	A → C; G → T	AUG → CUU	p.Met45Leu	missense_variant	Moderate
136 and 137	A → G; G → A	AGC → GAC	p.Ser46Asp	missense_variant	Moderate
141	A → C	CCA → CCC	p.Pro47 =	synonymous_variant	Low
143	G → A	AGA → AAA	p.Arg48Lys	missense_variant	Moderate
148	T → G	TTA → GTA	p.Leu50Val	missense_variant	Moderate

Note: CDS: Coding DNA Sequence (position within the coding region, not genomic coordinate) REF → ALT (nuc): Reference nucleotide (REF) changed to Alternate nucleotide (ALT). Codon Change: Triplet nucleotide (codon) before and after the mutation. HGVS.p: Human Genome Variation Society protein-level notation (e.g., p.Met45Leu = Methionine at position 45 changed to Leucine; “=“ means no change in amino acid). Effect (SnpEff): Functional classification of the variant (missense, synonymous, etc.). Impact: Predicted severity level by SnpEff (moderate indicated there are changes in the protein, but they do not always result in total loss of function, low indicated no changes.

**Table 4 animals-15-02781-t004:** Genetic diversity of ILEG among breeding villages.

No	Haplotype Position of ILEG
1	Hap_1: 1 [GR10]
2	Hap_2: 1 [AGI6]
3	Hap_3: 5 [AGI10 BW147; AGI4 AGI22 GR8]
4	Hap_4: 1 [AGI5]
5	Hap_5: 1 [BWB12]
6	Hap_6: 1 [AGI24]
7	Hap_7: 1 [LW2]
8	Hap_8: 1 [BW10]
9	Hap_9: 1 [GR2]
10	Hap_10: 1 [BW14]
11	Hap_11: 1 [AGI11]
12	Hap_12: 1 [AGI23]
13	Hap_13: 1 [BWC17]
14	Hap_14: 1 [AGI18]
15	Hap_15: 1 [AGI17]

**Table 5 animals-15-02781-t005:** Frequency of genetic group and average prolificacy from identified prolific data.

Gene	*n*	*Frequency of* *Genetic Group*			Genetic Group	No. of Does	Average Prolificacy
		AA	GA	GG	GG	11	1.89
BMPRIB	73	0.15	0.70	0.15	GA	51	1.60
		(11)	(51)	(11)	AA	11	1.49

**Table 6 animals-15-02781-t006:** Statistic description of prolificacy (offspring per kidding) by genetic group.

Genetic Group	n	Mean	Median	IQR	Shapiro–Wilk Normality Test	Kruskal–Wallis Test
					*p*-Value #	*p*-Value ##
GG	11	1.89 ± 0.35	2.00	0.00	0.0003	
GA	51	1.60 ± 0.52	1.50	1.00	0.0002	0.056
AA	11	1.49 ± 0.58	1.40	0.84	0.027	

Note: # *p* < 0.05 indicate that data is significantly deviated from normal distribution. ## *p* = 0.056 indicates statistical significance among GG, GA, and AA genetic groups.

**Table 7 animals-15-02781-t007:** The median comparison and pairwise test among three genetic groups.

Comparison	Median Differences (%)	Pairwise Mann–Whitney Tests (*p*-Value #)
GG vs. GA	+33.30	0.029
GG vs. AA	+42.90	0.040
GA vs. AA	+7.10	0.508

Note: # *p* < 0.05 indicate that data is significantly different.

**Table 8 animals-15-02781-t008:** Allele, genetic group frequency, and average prolificacy from none identified prolific data.

		Allele Frequency		Genetic Group Frequency			Genetic Group	No. of Does
Gene	*n*	A	G	AA	AG	GG	GG	53
BMPRIB	385	0.55	0.45	0.55	0.45	0.14	AG	316
				(16)	(316)	(53)	AA	16

**Table 9 animals-15-02781-t009:** Heterozygous, Homozygous for Allele G and A goats and their genetic group frequencies in several breeding villages of East Java.

Breeding Villages	*n*	Amount (Head)			Genetic Group Frequencies (%)		
		Heterozygous	Homozygous for Allele G	Homozygous for Allele A	GA	GG	AA
Malang I	38	36	2	0	94.74	5.26	0.00
Malang II	87	66	17	4	75.86	19.54	4.60
Blitar	22	19	3	0	86.36	13.64	0.00
Gresik	63	50	13	0	79.37	20.63	0.00
Lumajang	46	34	5	7	73.91	10.87	15.22
Nganjuk	32	26	6	0	81.25	18.75	0.00
Trenggalek	50	45	3	2	90.00	6.00	4.00
Banyuwangi	47	40	4	3	85.11	8.51	6.38
Total	385	316 ^b^	53 ^a^	16 ^a^			

Note: ^a,b^ different notations within the same row indicate a statistically significant difference.

## Data Availability

The original contributions presented in this study are included in the article. Further inquiries can be directed to the corresponding author.

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
