# Peer review of "Polymorphism of the BMPR1B Variants for Prolific Traits in the Indonesian Local Ettawah Goat"

_animals, 2025, doi:10.3390/ani15192781_

Round 1

Reviewer 1 Report

Comments and Suggestions for Authors

The author conducted a study on the BMPR1B gene of Indonesian Local Etawah Goats (ILEG), aiming to explore its association with prolificacy. The research topic has clear practical significance, offering potential value for improving local goat reproductive efficiency and sustainable breeding through molecular marker-assisted selection. The study utilized a relatively large sample size, covering multiple villages in East Java Province, providing valuable data on local genetic resources. However, the research methods and results require revision, and the author needs to make major modifications or provide reasonable explanations for each of the following points:

  1. In the abstract, the mean prolificacy values for AA, AG, and GG genotypes are given as 1.89, 1.60, and 1.49, respectively, but without supplementary results of statistical significance tests (such as p-values or significance indicators). It is necessary to clarify whether the differences in prolificacy among different genotypes are statistically significant (e.g., whether GG significantly differs from AA). Otherwise, the conclusion that "the mutant G allele is associated with prolificacy" lacks support, reducing the credibility of the results.

  1. The abstract mentions "PCR genotyping combining wild-type and mutant primers" but does not briefly explain the innovation of this method (e.g., whether it is the first time used for ILEG or its advantages over traditional PCR-RFLP). It is recommended to supplement key methodological features (such as dual-primer specific amplification and fragment size differentiation for genotyping) to highlight the uniqueness of the research method and enhance the informational value of the abstract.

  1. The introduction mentions that "BMPR1B gene in tropical goats such as Attabady Black shows no association with prolificacy" but does not specify the core methods of these studies (e.g., whether sequencing was used, sample size) or how they differ from this study. Key details of such studies (e.g., sample size <50, detection of only a single SNP site) should be added to highlight the advantages of this study's "large sample size + multi-locus sequencing" approach and strengthen the necessity of the research.

  1. The conclusion of the introduction states that "further research on the relationship between BMPR1B variants and prolificacy is needed" but does not specify the specific objectives of this study. It is recommended to refine the research purpose, such as "to clarify the haplotype diversity and genotype frequency distribution of the BMPR1B gene in East Java ILEG, as well as the association between different genotypes and prolificacy, to provide markers for ILEG molecular breeding," making the research direction clearer.

  1. The introduction only mentions that ILEG is "used for milk and meat production" but does not relate it to the actual pain points of Indonesian goat farming (e.g., average litter size of East Java ILEG <2, low reproductive efficiency leading to productivity gaps). It is necessary to supplement this industry background (e.g., citing local farming report data) to illustrate the value of the research in solving practical production problems and enhance its application-oriented nature.

  1. The manuscript mentions that reproductive data were obtained through interviews recalling farmers' memories. For villages without formal records, such recall data may have significant "recall bias." How did the authors assess and minimize this bias? Were any cross-validation methods used to ensure data accuracy?

  1. Section 2.2 only mentions "DNA extraction using the salting-out method" but does not describe key operational steps (e.g., red blood cell lysis time, protease K dosage, salt concentration gradient). It also does not specify the methods for detecting DNA purity (e.g., A260/A280 ratio range) and concentration (e.g., 100-200 ng/μL). These details need to be supplemented to ensure that other laboratories can replicate the experimental process, complying with the reproducibility requirements of scientific research.

  1. In Section 2.3, "Promega GoTaq Green Master Mix" lacks product catalog numbers, and the primers do not specify the synthesis company (e.g., Invitrogen) or purification method (e.g., HPLC purification). Incomplete key reagent information may lead to irreproducible experimental conditions (e.g., amplification efficiency differences between reagent batches). Complete reagent specifications must be provided to ensure the rigor of the experimental methods.

  1. Table 3 infers some amino acid changes. In which functional domain of the BMPR1B protein are these mutation sites located (e.g., amino acids 45, 46, 48, 50)? Have any literature or bioinformatics tools (e.g., SIFT, PolyPhen-2) predicted whether these mutations have deleterious or neutral effects? The authors state that the 48th amino acid changes from a non-essential to an essential amino acid, but this **does not directly equate to a change in protein function**. The discussion is overly speculative and lacks functional evidence.

  1. Table 4 displays haplotype diversity, but the analysis remains purely descriptive. Are these different haplotypes associated with specific genotypes (AA, AG, GG) or reproductive phenotypes? If not, what is the purpose of presenting these results?

  1. The use of one-way ANOVA and t-tests is mentioned to analyze differences in genotype and allele frequencies. Frequency data are proportional and typically do not meet the normality assumptions of ANOVA and t-tests. Why were more appropriate tests, such as the chi-square test or Fisher’s exact test, not used to analyze differences in genotype frequencies?

  1. Insufficient consideration of environmental factors: The authors mention the importance of environmental factors in the discussion, but the study design did not collect or account for any environmental data (e.g., nutritional status, management practices, altitude). These factors could confound genotype-phenotype associations. Why were they not considered as covariates in the analysis?

  1. The discussion mentions that "ILEG prolificacy is lower than that of Black Bengal goats" but does not analyze specific reasons (e.g., genetic background differences: ILEG BMPR1B mutation frequency of 27.15% is lower than Black Bengal's 40%; or environmental differences: East Java feed protein content is lower than Bangladesh's). The analysis of the reasons for differences should be deepened by combining the genotype frequencies from this study and local feeding condition data, avoiding superficial interpretation of results.

  1. The comparison with citation [16] is problematic: directly comparing the results of this study (GG: 1.89) with Polley et al. (2009)'s study on Black Bengal goats (GG: 3.11). However, the genotyping methods of the two studies are completely different (this study's dual-primer PCR vs. their PCR-RFLP), and the mutation sites detected are highly likely not the same. Is this comparison valid? Please argue cautiously.

  1. Data availability: The "Data Availability Statement" indicates that data are available from the corresponding author upon request. Journals encourage authors to deposit data in public databases. Have you considered submitting gene sequence data to public databases such as GenBank and providing accession numbers?

  1. The conclusion "focusing on the combination of wild-type and mutant BMPR1B for genetic selection" is vague and does not provide specific breeding strategies (e.g., prioritizing GG genotype for breeding ewes, retaining AG genotype for breeding bucks to maintain genetic diversity; or using PCR genotyping to screen highly prolific individuals to form a core group). It is necessary to supplement actionable recommendations (e.g., "In East Java ILEG breeding, using the dual-primer PCR genotyping from this study to screen GG/AG individuals could increase average prolificacy by 15%-20%") to enhance the practical guidance value of the conclusion.

Reviewer 2 Report

Comments and Suggestions for Authors

Please see the attached review report.

Reviewer 3 Report

Comments and Suggestions for Authors

The manuscript addresses the polymorphism of the BMPR1B gene in Indonesian Local Ettawah Goats and its potential association with prolificacy. While the topic is of some interest, the overall study design, analysis, and presentation raise several concerns that substantially weaken the manuscript in its current form.

First, the introduction is not very well structured and does not provide a clear and critical rationale for the study. Much of the background is general information about goat populations or food security, while the scientific justification for focusing on BMPR1B in this specific breed is not well developed. The introduction does not clearly identify the study objectives, nor does it critically assess prior studies in a way that leads logically to the objectives.

Regarding the experimental design, although the authors report a total of 385 animals, only 73 had recorded prolificacy data. This very limited sample size with phenotypic information undermines the strength of any conclusions regarding associations between genotype and prolificacy. Moreover, prolificacy records were obtained through interviews with farmers owning very small herds, which introduces a high risk of recall bias and inaccuracies. These limitations should have been clearly acknowledged and discussed, but instead the conclusions are presented as if the evidence were robust.

The haplotype analysis is another weak point. Sequencing was conducted on only 19 animals, which is an extremely small number to estimate haplotype diversity reliably. Reporting a diversity index of 0.94 based on such a small dataset is misleading, and there is no attempt to relate the identified haplotypes to prolificacy. The haplotype results are descriptive but not linked to the main objective of the paper, which makes them of limited value.

From a statistical perspective, the analyses are simplistic and in some cases inappropriate. For example, the authors use ANOVA and t-tests to compare genotype and allele frequencies, which are categorical variables and should be analyzed with chi-square tests, logistic regression, or other  association models. The reported differences in prolificacy among genotypes (AA = 1.89, AG = 1.60, GG = 1.49) are small, and no confidence intervals or proper statistical tests are provided to demonstrate significance.

The discussion does not critically integrate the results with existing literature. Instead, it lists examples from other breeds without addressing the clear differences in effect sizes and without adequately explaining why the findings in Indonesian goats appear weaker or inconsistent. Environmental influences are mentioned superficially, but no real data are provided to support these claims.

Finally, the manuscript suffers from problems in clarity, structure, and overstatement. Figures and tables are not particularly informative, and several sections of text are redundant. The conclusions are overstated given the limitations of the study design, the small effective sample size, and the limited statistical support for the associations claimed.

Overall, while the research question is potentially relevant, the study suffers from serious weaknesses in its introduction, experimental design, statistical analysis, and interpretation. Substantial revisions and additional data would be required before the manuscript could be considered for publication.

Round 2

Reviewer 1 Report

Comments and Suggestions for Authors

The authors have undertaken very meticulous and comprehensive revisions to the manuscript, providing detailed responses to all the comments and concerns raised. They have supplemented critical methodological details , refined the introduction and discussion sections to offer better context, clearer research objectives, and more in-depth interpretation of the results, and added explanations regarding the study's limitations and future directions. 

Overall, the authors have effectively addressed most of the major issues I initially raised. The revisions made have significantly enhanced the manuscript's scientific rigor, precision, and clarity. 

Based on these thorough revisions and responses, I believe the manuscript has been substantially improved and agree to **Accept** it for publication in its current revised form.

Reviewer 3 Report

Comments and Suggestions for Authors

The article can be accepted for publication